# Genomic Characterization of Serotype III/ST-17 Group B *Streptococcus* Strains with Antimicrobial Resistance Using Whole Genome Sequencing

**DOI:** 10.3390/biomedicines9101477

**Published:** 2021-10-15

**Authors:** Jen-Fu Hsu, Ming-Horng Tsai, Lee-Chung Lin, Shih-Ming Chu, Mei-Yin Lai, Hsuan-Rong Huang, Ming-Chou Chiang, Peng-Hong Yang, Jang-Jih Lu

**Affiliations:** 1Division of Pediatric Neonatology, Department of Pediatrics, Chang Gung Memorial Hospital, Taoyuan 333, Taiwan; jeff0724@gmail.com (J.-F.H.); kz6479@cgmh.org.tw (S.-M.C.); lmi818@msn.com (M.-Y.L.); qbonbon@gmail.com (H.-R.H.); cmc123@cgmh.org.tw (M.-C.C.); ph6619@cgmh.org.tw (P.-H.Y.); 2College of Medicine, Chang Gung University, Taoyuan 333, Taiwan; mingmin.tw@yahoo.com.tw; 3Division of Neonatology and Pediatric Hematology/Oncology, Department of Pediatrics, Chang Gung Memorial Hospital, Taoyuan 638, Taiwan; 4Department of Laboratory Medicine, Chang Gung Memorial Hospital at Linkou, Taoyuan 333, Taiwan; leollc@gmail.com; 5Department of Medical Biotechnology and Laboratory Science, Chang Gung University, Taoyuan 333, Taiwan

**Keywords:** group B streptococcus, next-generation sequencing, antimicrobial resistance, bloodstream infection, invasive disease

## Abstract

**Background:** Antibiotic-resistant type III/ST-17 *Streptococcus agalactiae* (group B *Streptococcus*, GBS) strain is predominant in neonatal invasive GBS diseases. We aimed to investigate the antibiotic resistance profiles and genetic characteristics of type III/ST-17 GBS strains. **Methods:** A total of 681 non-duplicate GBS isolates were typed (MLST, capsular types) and their antibiotic resistances were performed. Several molecular methods (WGS, PCR, sequencing and sequence analysis) were used to determine the genetic context of antibiotic resistant genes and pili genes. **Results:** The antibiotic resistant rates were significantly higher in type Ib (90.1%) and type III (71.1%) GBS isolates. WGS revealed that the loss of PI-1 genes and absence of ISSag5 was found in antibiotic-resistant III/ST-17 GBS isolates, which is replaced by a ~75-kb integrative and conjugative element, ICE*Sag37*, comprising multiple antibiotic resistance and virulence genes. Among 190 serotype III GBS isolates, the most common pilus island was PI-2b (58.4%) alone, which was found in 81.3% of the III/ST-17 GBS isolates. Loss of PI-1 and IS*Sag5* was significantly associated with antibiotic resistance (95.5% vs. 27.8%, *p* < 0.001). The presence of ICE*Sag37* was found in 83.6% of all III/ST-17 GBS isolates and 99.1% (105/106) of the antibiotic-resistant III/ST-17 GBS isolates. **Conclusions:** Loss of PI-1 and IS*Sag5*, which is replaced by ICE*Sag37* carrying multiple antibiotic resistance genes, accounts for the high antibiotic resistance rate in III/ST-17 GBS isolates. The emerging clonal expansion of this hypervirulent strain with antibiotic resistance after acquisition of ICE*Sag37* highlights the urgent need for continuous surveillance of GBS infections.

## 1. Introduction

The opportunistic pathogen group B *Streptococcus* (GBS, also known as *Streptococcus agalactiae*) is a commensal bacterium of the human genitourinary and digestive tracts and colonizes in 20–30% of pregnant women [1,2]. GBS has been reported as an important pathogen since the 1960s and infection by GBS is characterized by life-threatening sepsis, pneumonia and meningitis in young infants and immunocompromised adults [3,4,5]. A current universal screening strategy and administration of intrapartum antibiotic prophylaxis has decreased the incidence of GBS early-onset disease (day 0–6), but late-onset GBS infections (day 7–89) remain a significant burden worldwide [6,7,8,9].

GBS isolates express the antigenically unique capsular polysaccharide and can be categorized into 10 serotypes (Ia, Ib, and II-IX) [3,4,5,6,7,8]. Most clinical isolates and human carriage strains can be clustered into several major clonal complexes (CCs) by multilocus sequence typing (MLST), with type III/CC17 being the most virulent strain that causes the majority of neonatal meningitis and late-onset diseases [3,4,5,6,7,8]. The high virulence and invasiveness of the III/CC17 GBS strain may result from some specific genes encoding secreted and surface proteins, such as HvgA and pilus islands (PIs), which have been documented to mediate interactions between GBS and host cells [10,11,12]. Recent genome-based studies found that many CC17/III GBS strains displayed a high degree of genetic homology and spread of the specific clone has been reported [13,14,15].

GBS is uniformly susceptible to ß-lactams, but clindamycin and erythromycin are often used in penicillin-allergic patients. Recently, an increasing trend of resistance to erythromycin and clindamycin has been reported worldwide [13,15,16,17,18]. Expansion of certain GBS clones, such as the CC1 GBS among adults and CC17 GBS in neonates, was documented to be associated with the acquisition of resistance to macrolide and lincosamide antibiotics [19,20,21]. Another study provided evidence that acquisition of mobile genetic elements (MGEs) carrying antibiotic resistance genes in CC17 GBS strains is associated with the loss of pilus islands [22]. However, the actual mechanism by which antibiotic resistance genes interact with pilus structures remains unknown. In this study, we performed whole-genome sequencing (WGS) to elucidate the genetic basis of antibiotic resistance in ST17/III GBS strains.

## 2. Materials and Methods

### 2.1. Isolates Collection and Growth Conditions

A total of 681 nonduplicate GBS isolates were recovered from neonatal and adult patients (*n* = 182 and *n* = 399, respectively) with invasive GBS diseases who were admitted to Chang Gung Memorial Hospital (CGMH) between January 2005 and December 2018. Another 100 GBS isolates collected from colonized pregnant women between 2014 and 2015 were also included. All these isolates were preserved in sterile skim milk with 15% glycerol at −70 °C for long-term storage before they were cultured on blood agar plates and incubated at 37 °C in 5% CO_2_. This study was approved by the ethics committee of CGMH and written informed consent was provided by the pregnant women who had GBS colonization. For patients with GBS invasive diseases, a waiver of informed consent for anonymous data collection was approved.

### 2.2. Whole-Genome Sequencing

An invasive III/ST-17 GBS strain N5 was arbitrarily chosen as the reference strain for genomic comparisons. The genome of N5 was sequenced by WGS, which was performed using both PacBio^TM^ SMRT (Pacific Biosciences, Menlo Park, CA, USA) [23] and MiSeq^TM^ (Illumina, San Diego, CA, USA) [24] sequencing technologies. The sequencing library was prepared using a TruSeq DNA LT Sample Prep Kit (Illumina, San Diego, CA, USA) for the Illumina MiSeq system. Genomic libraries were generated using Nextera XT kits (Illumina, San Diego, CA, USA). The genome assembly was completely concordant with full-length perfectly aligning Illumina short reads. All the sequencing processes were performed using a DNA sequencing kit 4.02v2 (QIAGEN, Hilden, Germany) and SMRT cell 8 Pac (PacBio, Menlo Park, CA, USA). The Circlator tool (v1.4.0) was used to correct and linearize the genome, and QUAST (v4.5) was applied to evaluate the assembled genome quality. Other GBS strains including A28, N48, N96, P103 and P65 were used to generate contigs by the MiSeq^TM^ sequencing method. All genome sequences were blasted against the NCBI genome database to search for possible plasmid sequences. After the de novo assembled genome was generated, Prokka (v1,12) [25] was used for genome annotation and identification of rRNA-encoding and tRNA-encoding regions.

### 2.3. MLST, Serotyping, Pilus Typing and Antimicrobial Susceptibility Test

MLST was performed to evaluate all GBS isolates, and seven housekeeping genes, as previously described [26], were sequenced. Briefly, PCR fragments for seven housekeeping genes (*adhP*, *atr*, *glcK*, *glnA*, *pheS*, *sdhA* and *tkt*) were amplified and sequenced. The sequence type (ST) was determined via the *Streptococcus agalactiae* MLST database (http://pubmist.org/sagalactiae (accessed on 9 September 2021)). Sequence types not previously described were submitted to and were assigned by the *S. agalactiae* MLST database. The STs were grouped via the eBURST program into clonal complexes (CCs) whose members shared at least five of the seven MLST loci [27]; otherwise, an ST was considered a singleton.

The capsule genotypes were analyzed using the polymerase chain reaction (PCR) approach, and this assay, as well as the DNA isolation method, was described in our previous publication [28,29]. Pilus island content was confirmed by the multiplex PCR method as previously described [30]. Briefly, we used the primers provided by a recent study [30] for identification of the pilus Island (PI) marker, and a multiplex PCR assay was performed to analyze the distribution of GBS PI genes. The target genes, including alcohol dehydrogenase gbs0054 (*adhP*) as housekeeping loci, *sag*647 for PI-1, *sag*1406 for PI-2a and *san*1517 for PI-2b, were used to check in all of the type III GBS isolates. In addition, loss of PI-1 gene was confirmed by a set of primers that amplification of the regions flanking the PI operon. Then, primers were designed based on the sequence of ISSag5, three pili genes, ICE*Sag37* and the genes between the beginning of ISSag5 and end of PI-1 class C sortase (Table 1). The same PCR condition of Khodaei et al. [30] was used, and the PCR products were sent for sequencing (Bioneer OligoNucleotide Synthesis Co., Seoul, Korea).

Antibiotic susceptibility testing (AST) was performed by the disc diffusion method according to the guidelines of the Clinical and Laboratory Standards Institutes (CLSI) [31]. All GBS isolates were tested against penicillin, erythromycin, clindamycin, ampicillin, cefotaxime, vancomycin and teicoplanin. The minimum inhibitory concentrations of erythromycin and clindamycin were determined using the macrodilution method according to CLSI guidelines [31]. The detection of inducible clindamycin-resistant GBS (D-shape) was performed according to standard method [11,14].

**Table 1 biomedicines-09-01477-t001:** Principal oligonucleotide primers used in this study.

Gene	Sequence (5′−3′)	Product Size (bp)	Reference (DOI)
*ISSag5*	F: CAACAGATGCATCTCATTCTAATC	1459	this study
	R: TTCCTGCACATCTCAACTAA		
*PI-1*	F: CAAGATTGACCGGGTGGAGA	325	10.1016/j.micpath.2018.01.035 [30]
	R: ATGGGCAGTTAGAACGGCAT		
*PI-2a*	F: CGGGGTGCAAGTCAATAAGG	264	10.1016/j.micpath.2018.01.035 [30]
	R: GGAGCAGGGCATTTAGAAGGT		
*PI-2b*	F: CTCTGCTACCACCAAAGCGT	665	10.1016/j.micpath.2018.01.035 [30]
	R: GTGGGGGTAGGCTTAATGGC		
*ICESag37 head*	F: ACATAGCCCCGTCAGTATG	816	this study
	R: ATCACGTGGAGTGGTAGTG		
*ICESag37 tail*	F: GCAACGTGGTGAATTGATAGGG	1011	10.3389/fmicb.201.7.01921 [32]
	R: AAAACTGCACGATCAAACTCCG		

ICE: integrative and conjugative element.

## 3. Results

### 3.1. Serotyping and Antimicrobial Susceptibility Testing

Among the 681 GBS isolates, serotype III (*n* = 190) and VI (*n* = 191) were the most predominant. In neonates with invasive diseases, serotype III GBS (*n* = 118, 64.8%) was the most predominant, followed by type Ia (*n* = 33, 18.1%) and type Ib (*n* = 15, 8.2%). Type VI GBS (*n* = 150, 37.6%) was the most common strain in adult patients with invasive diseases, followed by type V (*n* = 59, 14.8%), type Ib (*n* = 56, 14.0%) and type III (*n* = 40, 10.0%). For pregnant women, the most common colonizing strains were type VI (*n* = 35) and type III (*n* = 25). A total of 190 (27.9%) type III GBS isolates were confirmed in this cohort, and 67.4% of them (*n* = 128) were ST-17/III GBS isolates. The phylogeny of the 190 type III GBS isolates, their sequence types and clonal complexes are presented in Figure 1. Based on the MLST analyses, most of the type III GBS isolates from neonatal invasive diseases were CC17/III, whereas those from adults were non-CC17/III type.

The antimicrobial susceptibility testing results are summarized in Table 2. All GBS isolates were susceptible to ampicillin, penicillin, vancomycin, teicoplanin and cefotaxime. Of the 681 GBS isolates, the antibiotic resistance to erythromycin and clindamycin was 49.5% and 48.9%, respectively. Most of the GBS isolates that were erythromycin resistant (*n* = 337) were also clindamycin resistant (95.0%, 320 isolates, *p* < 0.001 by Pearson χ^2^ test). For the 17 GBS isolates with resistance to erythromycin but sensitive to clindamycin, all the D-shape tests were negative. As we observed antimicrobial resistance profiles by serotype and ST, significantly higher rates of erythromycin and clindamycin resistance were noted in serotypes Ib (90.1%), III (72.6%) and V (67.9%). In addition, GBS strains that caused neonatal invasive diseases had a significantly higher antibiotic resistance rate (65.9% and 68.1%) than that which caused adult invasive diseases and that from colonized pregnant women (both *p* < 0.001), which mainly resulted from the high antibiotic resistance rates of 77.1−100% in serotype Ib and III GBS isolates.

### 3.2. Whole Genome Sequencing

Because serotype III has emerged as the most important strain in neonatal invasive diseases and the increasing trend of antimicrobial resistance [4], this study focused on the serotype III GBS isolates. We performed whole-genome sequencing (WGS) using a total of six ST17/III GBS isolates (N5, N48, P103, N96, A28 and P65) based on their sources, antibiotic resistance profiles, genes of antimicrobial resistance and patient demographics (Table 3). Strain NGBS128 and B105, both being ST17/III GBS isolates from neonatal with sepsis, had genomes of 2,074,179 bp and 2,273,717 bp, respectively (Figure 2). We investigated all the genes related to the component systems CovS/R, antibiotic resistance, pilus formation, capsular serotype and virulence. To track possible genomic clues regarding the mobile elements and insertion sequences (IS), comparative genome analyses were performed for all six GBS strains and the two reference strains.

We found that the genes of component systems CovS/R, capsular serotypes, and most virulence genes were not significantly different between antibiotic-susceptible (N96, A28, and P65) and antibiotic-resistant (N5, N48, and P103) GBS strains. The genes of pilus islands and nearby insertion sequences were significantly different between antibiotic-susceptible and resistant strains, and neither the PI-1 backbone protein nor the PI-1 ancillary protein was found in the three antibiotic resistant strains. After searching the ICEfinder database (https://db-mml.sjtu.edu.cn/ICEfinder/ICEfinder.html (accessed on 9 September 2021)), the presence of the insertion sequence IS*Sag5* was found upstream of the pili genes in all three antibiotic-susceptible GBS strains, but was not found in all three antibiotic-resistant GBS strains. In addition, one integrative and conjugative element, ICE*Sag37*, carrying multiple antibiotic resistant genes and virulence genes was found to replace the original region of PI-1 genes and ISSag5 in three antibiotic-resistant III/ST-17 GBS isolates, the Sag37 strain (type Ib/ST12) and the B105 reference strain (Figure 3).

The components and characterization of ICE*Sag37* were based on a recent publication [32]. ICE*Sag37* was first discovered by Zhou et al. in a type Ib/ST12 GBS strain isolated from blood samples of neonates with bacteremia [32]. ICESag37 has the size of 73,429 kb and contains genes of virulence, a two-component signal transduction system (*nisK*/*nisR*), integrase, genes of MobC and relaxases and resistance genes to multiple antibiotics, including *ermB* (erythromycin), *tetO* (tetracycline), *aadE*, *aphA* and *ant-6* (aminoglycosides). IS*Sag12* was integrated into the mosaic area of antibiotic resistance genes of ICE*Sag37*, as shown in Figure 3.

PCR was performed in all six ST-17/III GBS isolates to verify the results and confirm the relationship between ISSag5 and PI-1 genes. We found loss of PI-1 and replacement by ICE*Sag37* in all three antibiotic-resistant ST-17/III GBS isolates, as shown in Figure 4, which are compatible with the findings of WGS.

### 3.3. Pilus Genes, ISSag5 and ICESag37 in All Serotype III GBS Isolates

A total of 190 serotype III GBS isolates were used to verify the results and the presence of IS*Sag5*, pilus island genes and ICE*Sag37* and antibiotic susceptibility patterns are summarized in Table 4. Among all 190 type III GBS isolates, the most common pilus island was PI-2b alone (58.4%, *n* = 111), followed by PI-1 plus PI-2a (26.3%, *n* = 50) and PI-1 plus PI-2b (15.3%, *n* = 29). A significant association was found between the PI genes and antibiotic resistance profiles (loss of PI-1 was significantly associated with antibiotic resistance, 95.5% vs. 27.8%, *p* < 0.001), and all the type III GBS isolates without PI-1 were found to lack IS*Sag5*. The sensitivity and specificity of loss of PI-1 and the absence of IS*Sag5* to predict antibiotic resistance to both erythromycin and clindamycin were 95.5% and 82.8%, respectively. For 104 ST-17/serotype III GBS isolates without ISSag5 and loss of PI-1, 103 (99.0%) were resistant to both erythromycin and clindamycin. In addition, all serotype III/ST-17 isolates had PI-2b.

We found that loss of PI-1, as well as negative ISSag5, was significantly associated with the presence of ICE*Sag37* (Table 4). Among 111 type III GBS isolates without PI-1 and ISSag5, a total of 94.6% (*n* = 105) had ICE*Sag37*. In addition, the presence of ICE*Sag37* was found in 83.6% (107/128) of all III/ST-17 GBS isolates and 99.1% (105/106) of the antibiotic-resistant III/ST-17 GBS isolates.

## 4. Discussion

Our previous studies have found an increasing trend of the type III/ST-17 GBS strain with a high antibiotic resistance rate that caused GBS neonatal infections [4,33], which is compatible with the changing epidemiology of GBS susceptibility profiles in the literature [15,34,35]. In our cohort, we found that the emerging antibiotic-resistant III/ST-17 GBS isolates resulted from the acquisition of ICE*Sag37*, comprising multiple antibiotic resistance genes, in a specific genomic locus of nearly all clinical GBS isolates and balanced the loss of PI-1 and IS*Sag5*. Although these isolates came from a clinical collection of more than 10 years, the antibiotic-resistant ST17/III GBS isolates gradually predominated over non-ST17 type III GBS isolates in our institute [33]. In addition, our genome analysis is consistent with other comparative genomics and phylogenetic analyses that the isolates of the GBS CC17 population have a high degree of genetic homogeneity and a low rate of genetic recombination [15,19,22].

The prevalence of antibiotic resistance to clindamycin and erythromycin varies greatly in different geographic areas, with studies from Asia reporting a much higher resistance rate than Western countries [35,36]. The GBS resistance to erythromycin and clindamycin was 26.7–46% and 22.1–47%, respectively, with an upward trend [35,36,37]. No specific relationship between antibiotic resistance and specific serotypes was found in previous studies [30,34,35,36]. In our institute, serotype Ib and III GBS isolates had a significantly higher percentage of antibiotic resistance than other serotypes, which is notably different from previous studies [30,34]. Recent reports also found that the resistance of GBS isolates to other antibiotic classes, such as aminoglycosides and fluoroquinolones, is starting to rise [38,39]. Because continuing trends of increased GBS resistance may lead to the necessity of revised therapeutic options or prophylaxis, large-scale and long-term strain surveillance may be necessary to provide updated information.

All GBS strains carry at least one of the three pilus variants (types 1, 2a and 2b). Pili were originally documented to play an important role in cell adhesion, transcytosis and enhanced penetration of the blood–brain barrier, which can lead to sepsis and meningitis [40,41]. The presence of particular GBS pilus type profiles was found to be associated with different serotypes and phylogenetic lineages [42,43,44]. For example, most CC17/III GBS strains have a combination of PI-1 and PI-2b genes, which is relatively uncommon in other clonal complexes [42,43,44]. However, the loss of PI-1, found in 58.4% of our type III GBS strains, is not uncommon in other GBS serotypes but has only been described three times in CC17/III GBS isolates [15,22,44] and among bovine isolates [22]. Both Campisi et al. and Teatero et al. found that the CC17/III GBS strains devoid of PI-1 had an ICE or MGE encoding antibiotic resistance that was integrated simultaneously with the loss of PI-1 and replaced the original PI-1 at the same location on the chromosome [15,22]. This is similar to our finding that the integration of an MGE or an ICE encoding antibiotic resistance genes correlates with clonal expansion of the strains to predominate the geographic area.

An ICE contains a various group of MGEs and integrases to integrate into the host chromosome. The major effects of ICEs are carriages of several phenotypical genes and contribution of bacterial evolution and adaption after integration. ICE*Sag37* was first discovered by Zhou et al. [32] in the strain *S. agalactiae* Sag37, a type Ib/ST12 GBS isolate with high antibiotic resistance. The ICESag37 has been described to belong to the ICE*Sa2603* family-like ICE because of homology and high sequence similarity to a ~54 kb ICE from *S. agalactaie* 2603V/R strain [32,45]. In addition, the antibiotic resistance gene cluster of ICESag37 was found to be homologous to a plasmid in *E. faecalis*, which indicates the possibility of inter-species genetic exchange [32]. Although several ICEs have been reported in *S. agalactiae*, this is the first to find an original type Ib/ST12 ICESag37 predominant in type III/ST17 GBS isolates. The component genes involved in recombination and mobility can be found in the ICESag37, and the conserved gene, *rum*A, has been found to be a hotspot for the integration of streptococcal ICEs [32,46].

Insertion sequences (ISs) are a kind of intracellular MGEs or transposable elements that have an important impact on genome structure and function [47]. Various ISs have been reported in GBS strains [48,49,50], some of which cause genetic alterations or frameshift mutations that change the virulence or invasiveness of the bacteria. To our knowledge, this is the first study to find a significant relationship of negative ISSag5 and loss of PI-1. In contrast, IS*Sag12* was integrated into the mosaic antibiotic genes of ICE*Sag37* [32], although the source was unknown and the function was not investigated. In all antibiotic-resistant GBS strains, the absence of ISSag5 was confirmed by the amplification of a fragment of about 450 bp corresponding to the empty intergenic region. Although loss of PI-1 was found in all of our type III GBS strains with absence of ISSag5, the significant association was not 100% in other GBS serotypes (data not shown).

## 5. Conclusions

In summary, we report here the antibiotic resistance profiles of clinical GBS isolates and colonized GBS isolates and provide data showing the increasing antibiotic resistance rate of neonatal invasive strains mainly from the hypervirulent CC17/III GBS. The PI-1 loss, as well as the absence of ISSag5, were replaced by the ICESag37 carrying multiple antibiotic resistance genes, accounting for the high antibiotic resistance rate of type III/ST17 GBS strains in our cohort. This clonal expansion highlights the necessity of continuous surveillance of GBS infections. In addition, the application of WGS data analyses of specific strain samples to investigate genetic contents was validated in our study, which may provide novel information and be useful for public health applications.

## Figures and Tables

**Figure 1 biomedicines-09-01477-f001:**
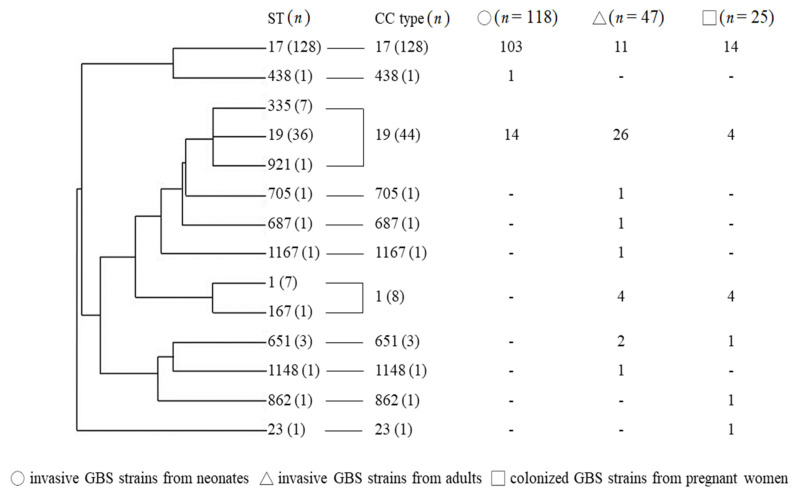
Phylogeny of the 190 type III GBS isolates from neonates and adults with invasive GBS diseases and colonized pregnant women. Distributions of sequence types and clonal complexes from different sources are presented; - means zero.

**Figure 2 biomedicines-09-01477-f002:**
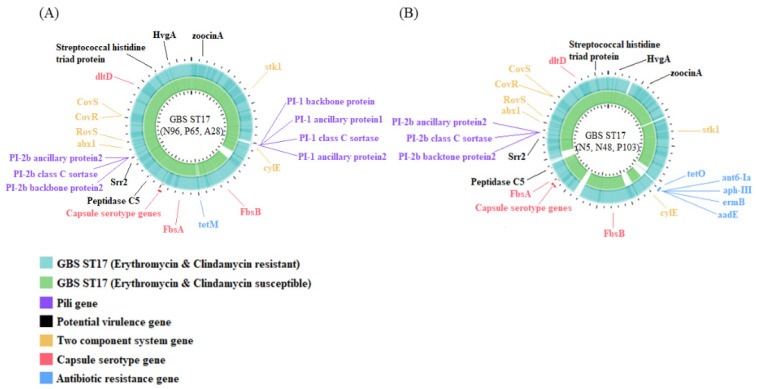
Whole-genome analysis of antibiotic susceptible (N5, N48, P103) and resistant (N96, P65, A28) serotype III/ST17 GBS strains. Genome scale in mega base pairs of these two reference strains are given in the innermost circle, respectively. TBLASTN comparisons of the reference NGBS128 (CP012480) genome (**A**) and B105 (CP021773.1) genome (**B**) to the complete genomes of antibiotic resistant and susceptible III/ST-17 GBS strains are shown in different colors. The genomes of two component systems CovS/R and genes related to capsular serotypes, pili, virulence and antibiotic resistance are presented in different colors.

**Figure 3 biomedicines-09-01477-f003:**
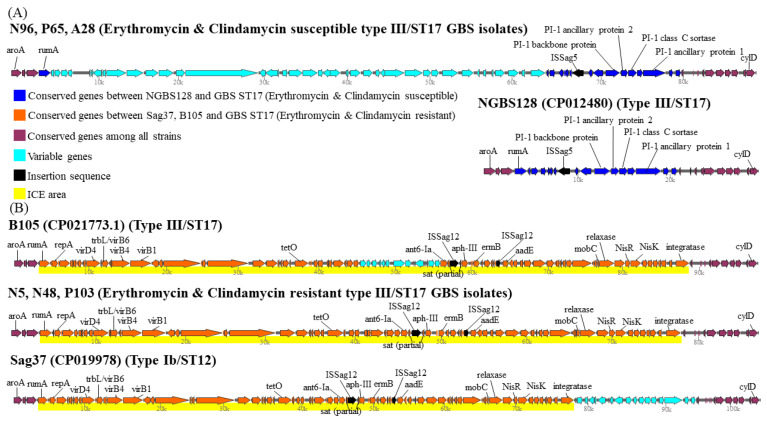
Pilus Island 1 is a hotspot for integration of integrative and conjugative element, ICE*Sag37*, carrying antimicrobial genes and virulence genes. (**A**) The top schematic shows the region between the beginning of gene aroA and the end of gene cylD (from nucleotides 589,950 to 619,400) in the genome of three antimicrobial susceptible strains (N96, P65, A28) and comparison with that depicted in the second schematic for the region of reference strain NGBS128. This region contains several PI-1 genes (blue) and ISsag5. (**B**) Loss of PI-1 and ISSag5 is replaced by acquisition of an integrative and conjugative element (ICE), ICESag37, carrying multiple resistance genes and virulence genes in three antimicrobial resistant ST-17/III GBS strains, the reference strain B105 and Sag37 (type Ib/ST12). The bottom schematic show that in these antimicrobial resistant GBS strains, the site-specific integration of ICESag37, starting from the beginning of rumA to the end of integratase, is clearly visible and shown in yellow color. Conserved genes are indicated in purple, variable genes are indicated in light blue. Antimicrobial resistance genes are indicated.

**Figure 4 biomedicines-09-01477-f004:**
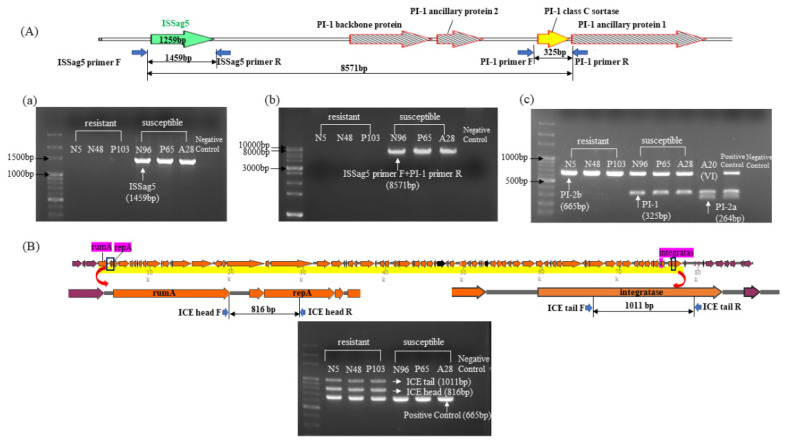
(**A**) The schematic shows the location and relationship between ISSag5 and several PI-1 genes, and the PCR results of several genes. (**a**) PCR results of the ISSag5 (1459 bp) using three antibiotic susceptible and three antibiotic resistant GBS strains by designed primer F and primer R. The ISSag5 sequence is based on searching from NCBI, and documented in all three antibiotic susceptible strains (N96, A28 and P65). (**b**) PCR results of genes from the beginning of ISSag5 to the end of PI-1 class C sortase (8571 bp) by designed primer F and PI-1 primer R. (**c**) PCR results of PI-1, PI-2a and PI-2b genes in antibiotic-susceptible and antibiotic-resistant III/ST-17 GBS strains. (**B**) The schematic shows the beginning of ICESag37 from *rum*A to the end of ICESag37 at integratase. The primers were designed for ICESag37 head and ICESag37 tail, respectively. The multiplex PCR was performed in three antibiotic-resistant and three antibiotic-susceptible III/ST-17 GBS isolates for verification.

**Table 2 biomedicines-09-01477-t002:** Serotype distributions and antibiotic resistance of 681 GBS isolates from maternal colonization, neonatal and adult invasive diseases.

No. (%) of Resistant GBS Isolates								
Sources	Maternal Colonized Strains (Total *n* = 100)		Adult Invasive Strains (Total *n* = 399)		Neonatal Invasive Strains (Total *n* = 182)		Total (*n* = 681 GBS Isolates)	
Serotype(*n*)/ Antibiotics	Erythromycin	Clindamycin	Erythromycin	Clindamycin	Erythromycin	Clindamycin	Erythromycin	Clindamycin
Ia (*n* = 80)	1/7 (14.3)	1/7 (14.3)	10/40 (25.0)	7/40 (17.5)	8/33 (24.2)	10/33 (30.3)	19/80 (23.8)	18/80 (22.5)
Ib (*n* = 81)	8/10 (80.0)	8/10 (80.0)	50/56 (89.3)	50/56 (89.3)	15/15 (100.0)	15/15 (100.0)	73/81 (90.1)	73/81 (90.1)
II (*n* = 47)	1/6 (16.7)	1/6 (16.7)	5/38 (13.2)	5/38 (13.2)	0/3 (0)	0/3 (0)	6/47 (12.8)	6/47 (12.8)
III (*n* = 190)	16/25 (64.0)	15/25 (60.0)	25/47 (53.2)	29/47 (61.7)	97/118 (82.2)	91/18 (77.1)	138/190 (72.6)	135/190 (71.1)
V (*n* = 81)	11/16 (68.8)	11/16 (68.8)	40/59 (67.8)	39/59 (66.1)	4/6 (66.7)	4/6 (66.7)	55/81 (67.9)	54/81 (66.7)
VI (*n* = 192)	6/35 (17.1)	6/35 (17.1)	40/150 (26.7)	40/150 (26.7)	0/7 (0)	0/7 (0)	46/192 (24.0)	46/192 (24.0)
Others (10)	0/1 (0)	0/1 (0)	0/9 (0)	1/9 (11.1)	0/0 (0)	0/0 (0)	0/10 (0)	1/10 (10.0)
All (*n* = 681)	43/100 (43.0)	42/100 (42.0)	170/399 (42.6)	171/399 (42.9)	124/182 (68.1)	120/182 (65.9)	337/681 (49.5)	333/681 (48.9)

All 681 GBS isolates are susceptible to vancomycin, ampicillin, penicillin, cefotaxime, and teicoplanin. All data are expressed as resistant strains/total strains (percentage).

**Table 3 biomedicines-09-01477-t003:** Information for the six serotype III/ST-17 group B streptococcus (GBS) strains.

Strain	Source	Age/Gender	Resistance Genes	Antimicrobial Resistance Profiles	
				Erythromycin	Clindamycin
N5	cerebrospinal fluid	newborn/M	*tetM*, *tetO*, *ermB*, *ant6-Ia*, *aphIII*, *aadE*	resistance	resistance
N48	cerebrospinal fluid	newborn/F	*tetO*, *ermB*, *ant6-Ia*, *aadE*, *aphIII*	resistance	resistance
P103	swab	pregnant women/F	*tetM*, *tetO*, *ermB*, *ant6-Ia*, *aphIII*, *aadE*	resistance	resistance
N96	cerebrospinal fluid	newborn/M	*tetM*	sensitive	sensitive
A28	blood	adult/M	*tetM*	sensitive	sensitive
P65	swab	pregnant women/F	*tetM*	sensitive	sensitive

**Table 4 biomedicines-09-01477-t004:** The antibiotic susceptibility patterns, pilus distribution and presence of IS*Sag5* in all serotype III GBS isolates.

No. (%) of All 190 Type III GBS Isolates						
IS*Sag* 5	Presence of IS*Sag* 5 (Total *n* = 33, 17.4%)			Absence of IS*Sag* 5 (Total *n* = 157, 82.6%)		
Pilus genes	PI-1 + PI-2a	PI-1 + PI-2b	PI-2b	PI-1 + PI-2a	PI-1 + PI-2b	PI-2b
Total *n* (%)	4 (2.1%)	29 (15.3%)	0 (0)	46 (24.2%)	0 (0)	111 (58.4%)
Sequence types						
ST-17 (total *n* = 128)	0 (0)	24 (82.8)	-	0 (0)	-	104 (93.7)
Non-ST17 (total *n* = 62)	4 (100)	5 (17.2)	-	46 (100)	-	7 (6.3)
Sources						
Neonatal invasive diseases	1 (25.0)	19 (65.5)	-	12 (26.1)	-	86 (77.5)
Adult invasive diseases	1 (25.0)	6 (20.7)	-	29 (63.0)	-	11 (9.9)
Maternal colonization	2 (50.0)	4 (13.8)	-	5 (10.9)	-	14 (12.6)
ICE*Sag37*						
Presence	0 (0)	3 (10.3)	-	8 (17.4)	-	105 (94.6)
Absence	4 (100)	26 (89.7)	-	38 (82.6)	-	6 (5.4)
Antibiotic resistance patterns (No. (%) of resistant GBS isolates)						
Ery (R) + Clin (R)	0 (0)	7 (24.1)	-	15 (32.6)	-	106 (95.5)
Ery (S) + Clin (S)	4 (100)	15 (51.7)	-	23 (50.0)	-	3 (2.7)
Ery (R) + Clin (S)	0 (0)	7 (24.1)	-	3 (6.5)	-	0 (0)
Ery (S) + Clin (R)	0 (0)	0 (0)	-	5 (10.9)	-	2 (1.8)

Ery: erythromycin; Clin: clindamycin; R: resistant; S: susceptible. All GBS isolates are susceptible to vancomycin, teicoplanin, ampicillin, penicillin, and cefotaxime; - means zero.

## Data Availability

Datasets used/or analyzed during the current study are available from the corresponding author on reasonable request.

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
