# Peer review of "Genomic Characterization of Serotype III/ST-17 Group B Streptococcus Strains with Antimicrobial Resistance Using Whole Genome Sequencing"

_biomedicines, 2021, doi:10.3390/biomedicines9101477_

Round 1
Reviewer 1 Report
The authors reported the antibiotic resistance profiles of clinical GBS isolates and colonized GBS isolates and provide data showing the increasing antibiotic resistance rate of neonatal invasive strains , mainly from the CC17/ IIIGBS . The novel information that could be useful for public health authorities were presented.
Author Response
Dear reviewer:
Please see the attachment, thank you.
Best regard,
Tsai Ming Horng

Reviewer 2 Report
The group aimed to investigate the antibiotic resistance profiles and genetic characteristics of type III/ST-17 GBS strains. They used state-of-the-art methods to achieve their aims. In my opinion, the continuous surveillance of GBS infection is extremely important worldwide. It is an excellent manuscript that connects basic science and the clinical significance of researches. However, the work could fit better in the scope of the other MDPI’s journals (Pathogens, Microorganisms).
I suggest correcting the following minor points:
Page 4, line 150 and 156: Consistent writing should be used to abbreviate significance (p or P)
Page 4, line 147: Ampicillin should be involved among the antibiotics in the text, because in the table legends it was mentioned (Table 2 and 4)
Page 6, line 15: In my opinion instead of N47, N48 should be used (I think N48 is the correct, as it was mentioned more times in the text)
Page 8, line 85: There is an unnecessary space in ISSag 5 (in the title of the Table 4)
Author Response

(The authors gave the same response as above.)
